# In-Pero: Exploiting Deep Learning Embeddings of Protein Sequences to Predict the Localisation of Peroxisomal Proteins

**DOI:** 10.3390/ijms22126409

**Published:** 2021-06-15

**Authors:** Marco Anteghini, Vitor Martins dos Santos, Edoardo Saccenti

**Affiliations:** 1Laboratory of Systems and Synthetic Biology, Wageningen University & Research, Stippeneng 4, 6708 WE Wageningen, The Netherlands; vitor.martinsdossantos@wur.nl; 2LifeGlimmer GmbH, 12163 Berlin, Germany

**Keywords:** protein sequence encoding and embedding, machine learning, neural networks, subcellular localisation, sub-peroxisomal localisation, sub-mitochondrial localisation

## Abstract

Peroxisomes are ubiquitous membrane-bound organelles, and aberrant localisation of peroxisomal proteins contributes to the pathogenesis of several disorders. Many computational methods focus on assigning protein sequences to subcellular compartments, but there are no specific tools tailored for the sub-localisation (matrix vs. membrane) of peroxisome proteins. We present here In-Pero, a new method for predicting protein sub-peroxisomal cellular localisation. In-Pero combines standard machine learning approaches with recently proposed multi-dimensional deep-learning representations of the protein amino-acid sequence. It showed a classification accuracy above 0.9 in predicting peroxisomal matrix and membrane proteins. The method is trained and tested using a double cross-validation approach on a curated data set comprising 160 peroxisomal proteins with experimental evidence for sub-peroxisomal localisation. We further show that the proposed approach can be easily adapted (In-Mito) to the prediction of mitochondrial protein localisation obtaining performances for certain classes of proteins (matrix and inner-membrane) superior to existing tools.

## 1. Introduction

In eukaryotes, there are ten main subcellular localisations which can be further subdivided into intra-organellar compartments (see Figure 1A). These organelles perform one or more, and often complementary, specific tasks in the cellular machinery. Examples of organelles are the nucleus, for the storage of genetic (DNA) material, mitochondria for the production of energy and the peroxisome.

The organelles provide suitable biological conditions for proteins and the correct transport of a protein to its final destination is crucial to its function. Failure in protein transport systems has been associated with several disorders including Alzheimer’s and cancers [1,2,3].

It has been observed that proteins from different organelles show signatures, in their amino acid composition, that associate with their subcellular localisation [4]. This has led to the hypothesis that each protein has evolved to function optimally in a given subcellular compartment, and to the idea that the information encoded in the sequence can be used to predict the subcellular localisation.

Since the pioneering work of Nakashima and Nishikawa, who used the amino acid composition to discriminate between intra- and extra-cellular proteins [5], several studies have been proposed to predict protein localisation (see [6] for comprehensive reviews). A list of the most common tools for subcellular localisation includes BaCello [7] a predictor based on different Support Vector Machines (SVM) organised in a decision tree; Phobius [8], a combined transmembrane topology and signal peptide predictor; WoLF PSORT [9] a *k*-nearest neighbors based classifier; TPpred3 [10], an SVM predictor exploiting N-terminal targeting peptides.

Nowadays, many bioinformatics methods for subcellular and sub-organelle localisation are easily findable and accessible [7,9,10,11]. Moreover, the recent applications of machine learning (ML) and deep-learning (DL) approaches to encode protein sequences, has shown promising results in several tasks, including subcellular classification [12,13,14,15,16,17,18].

In these approaches, protein sequences are commonly transformed to numerical representations that can be mathematically manipulated. Classically, these representations are referred to as “encodings” and can be broadly subdivided in four categories (*i*) binary encoding, (ii) encoding based on physical-chemical properties, (iii) evolution-based encoding and, (iv) structural encoding [19]. Examples are the one-hot encoding (1HOT) [19], the residue physical-chemical properties encoding (PROP) [20], the Position-Specific Scoring Matrix (PSSM) [21,22].

Recently, deep learning methods have also been proposed and applied to extracting fundamental features of a protein and to embed them into a statistical representation that is semantically rich and structurally, evolutionary, and bio-physically grounded [12]. These statistical representations are known as deep-learning embeddings (DL-embeddings) and are a multidimensional transformation of the protein sequence obtained using DL to extract and learn the information from the huge amount of protein sequences available in biological databases.

We can take advantage of these embeddings for several tasks, especially subcellular localisation [14]. Two of the most promising DL-embeddings are the Unified Representation (UniRep) [12] and the Sequence-to-Vector (SeqVec) [13] embeddings. UniRep [12] provides amino-acid embedding containing meaningful physicochemically and phylogenetic clusters and proved to be efficient for distinguishing proteins from various SCOP (structural classifications of proteins) classes. SeqVec showed similar results and optimal performance for predicting subcellular localisation, including peroxisomes [13].

Peroxisomes (see Figure 1B) are ubiquitous organelles surrounded by a single biomembrane that are relevant to many metabolic pathways like phospholipid biosynthesis, fatty acid beta-oxidation, bile acid synthesis, docosahexaenoic acid synthesis, fatty acid alpha-oxidation, glyoxylate metabolism, amino acid degradation, and ROS/RNS metabolism [23]. Peroxisomes are also involved in non-metabolic functions, like cellular stress responses, response to pathogens and antiviral defence, and cellular signalling [24]. Because of this they gained the appellative of "protective" organelles [24] and dysfunctions in peroxisomal proteins have been associated with metabolic disorders [23,24]. However, the full extent of their functions is still largely unknown [25] and the discovery of new peroxisomal proteins can facilitate further knowledge acquisition.

This leads to the problem of determining the localisation of peroxisome proteins. For instance, both membrane contact site (MCS) proteins [26] and peroxisomal transporters (PT) [27] are found on the membrane: that is, distinguishing between proteins located on the peroxisomal membrane or in its granular matrix is thus a fundamental step for the characterization of unknown peroxisomal proteins.

The problem of protein sub-peroxisomal localisation has received limited attention: as for today, the only way to retrieve information about the sub-peroxisomal localisation is to check for short conserved sequence motif known as signal motifs, or protein targeting signals (PTS) as implemented in the PeroxisomeDB server [28] (www.peroxisomedb.org, accessed on 1 June 2020). Through PeroxisomeDB, given a FASTA sequence as input, it is possible to identify PEX19BS, PTS1 and PTS2 targeting signals: more precisely, PTS1 and PTS2, can identify peroxisomal matrix proteins while PEX19BS can identify peroxisomal membrane proteins.

In this study, we address the problem of predicting the sub-localisation of peroxisomal protein using a computational strategy that combines protein-sequence embedding with classical machine learning. We reviewed and compared four different machine learning approaches, namely Logistic Regression (LR), Random Forest (RF), Support Vector Machine (SVM), and Partial Least Square Discriminant Analysis (PLS-DA) in combination with five protein embedding approaches: residue one-hot encoding (1HOT), residue physical-chemical properties (PROP), Position Specific Scoring Matrices (PSSM), Unified Representation, Sequence-to-Vector.

Based on our comparative study, we built a computational pipeline (In-Pero), which is based on Support Vector Machines and the combination of UniRep and SeqVec embedding. We also tested our approach for sub-mitochondrial localisation, obtaining a predictor (In-Mito) that outperformed most of the existing classifiers.

## 2. Results

### 2.1. Selection of the Best Classifier for Sub-Peroxisomal Prediction

We compared four commonly used machine learning approaches (Logistic Regression, Partial Least Squares Discriminant analysis, Random Forest and Support Vector Machines) in combination with different protein sequence encodings and embeddings to select the best classification strategy to predict the sub-localisation of peroxisomal proteins. Results are summarised, per classification algorithm, in Table 1 where different metrics for model quality quantification are given. All results were obtained with repeated double cross-validation to avoid model overfitting and bias.

In general, Logistic regression (Table 1a) and Support vector machines (Table 1b) showed similar performance, superior to PLS-DA Table 1c) and Random Forest (Table 1d). However, the prediction model built using SVM has a smaller standard deviation, indicating higher stability.

We observed that combining two different encodings and/or embeddings gives a better prediction of the peroxisomal sub-localisation. In particular, concatenating UniRep and SeqVec showed a noticeable improvement in the performances. That indicates that the two embeddings carry different and complementary information about the properties of the protein sequence, as given in Figure 2, that show how the two embeddings are not correlated.

### 2.2. In-PERO a Tool for the Prediction of Peroxisomal Protein Sub-Localisation

Based on the results obtained and discussed in Section 2.1, we developed In-Pero, a computational pipeline to predict the Intra-Peroxisomal localisation of a proximal protein, that is, to discriminate between matrix and membrane proteins.

In-Pero is based on a Support Vector Machine classifier trained on the statistical representation of protein sequences obtained by the combination of two deep-learning embeddings (UniRep + SecVec).

In-Pero consists of four main steps (see Figure 3A).

Input of the protein sequence in FASTA format.Calculation of the statistical representation of the protein sequence using the UniRep (1×1900) and the SeqVec (1×1024) embeddings.Merging of the two statistical representation to obtain a 2924-dimensional representation of the protein sequence.Prediction of the subcellular localization using the trained SVM prediction model.

In-Pero is implemented in Python and work in command line modality. An example of the input command line and output is given in Figure 3B.

### 2.3. Validation of Sub-Peroxisomal Membrane Protein Prediction

The In-Pero prediction tool was trained and validated using a double cross-validation strategy (see Section 4.7) with a stratified 5-fold splitting. The predictive capability of the model was assessed on the data that have not been used for model calibration (i.e., the selection of meta-parameters to obtain the best prediction quality). This approach is a proxy for the use of an external data set for experimental validation, and ensures unbiased model assessment and reduces the risk of over-fitting.

Despite all precautions, we believe it is important to benchmark In-Pero against existing tools. However, at the time of this writing, there are no existing computational specifically designed tools for the sub-localisation of peroxisomal protein. As a work-around, we compared the prediction of In-Pero with those of TMHMM server (see Section 4.9) using a set of 116 peroxisomal protein of unknown sub-peroxisomal localisation (see Section 4.2.3) which have not been used to train the In-Pero classifier.

When In-Pero is run on these 116 proteins, we obtained membrane localisation for 48 and matrix localisation for 68. We tested the 48 protein classified as membrane proteins using TMHMM: 7 were predicted as transmembrane proteins while 13 have characteristics compatible with transmembrane localisation (a value >= 1 for at least one among the ExpAA, First60 and PredHel scores, see Section 4.9). Prediction results are given in Table 2.

For two of the protein predicted as transmembrane proteins (064883 and P20138), there is experimental evidence of them being involved in to various cellular membranes  [29,30], while the sub-peroxisomal location is not reported. Among the others, four (Q84P23, Q84P17, Q84P21, and Q9M0X9) are present in *Arabidopsis thaliana* while P08659 is present in *Photinus pyralis*. For these five proteins, neither membrane localisation nor the sub-peroxisomal location are given, making them candidates for a more precise annotation.

It should be noted that TMHMM method focuses on predicting transmembrane regions, not predicting subcellular location. However, we can speculate that also peroxisome membrane protein may share some structural and physico-chemical properties similar to cell membrane proteins; thus confronting the results of TMHMM with In-Pero can provide an independent, albeit only partial, validation of the In-Pero classifier.

### 2.4. Extending In-Pero to Predict Sub-Mitochondrial Proteins

To explore further the applicability of the combination of machine learning and deep-learning protein sequence embeddings to other problems related to the prediction of protein localisation, we applied In-Pero for sub-mitochondrial classification.

Mitochondrial proteins are physiologically active in different compartments (the matrix, the internal membrane, the inter-membrane space and the external membrane) and their aberrant localisation contributes to the pathogenesis of human mitochondrial pathologies [31]. By adapting In-Pero to a multiclass classification problem, we obtained the In-Mito predictor. We considered both an SVM and a multinomial Logistic Regression as classification algorithms since, in this case, they performed similarly.

There are several tools available for the prediction of sub-mitochondrial localisation. We compared In-Mito against SubMitoPred [32], DeepMito [15], and DeepPred-SubMito [33].

We tested our model with the SM424-18 and SubMitoPred data sets (see Section 4.2.4 for more details). Results are given in Table 3.

In-Mito compared favourably to existing predictors, especially in respect to methods designed to classify all four mitochondrial compartments. Moreover, In-Mito shows a well-balanced capability to predict all four different compartments. In particular, In-Mito shows excellent performance in the prediction of matrix proteins and inner membrane proteins, which are the two most abundant subcellular compartments (80% of the SubMitoPred data set).

For this multi-class problem, we obtained better prediction performance using either logistic regression (for matrix protein) or SVM (for inter-membrane proteins). This supports the idea of possibly combining different predictors for better classification.

Given the accuracy of the classifications obtained with our approach, we also implemented the tool In-Mito for sub-mitochondrial classification, which works in the same way as In-Pero (Figure 3). In particular, the final output here consists of one among the four possible sub-mitochondrial compartments.

## 3. Discussion

With this work, we covered a less explored area of bioinformatics analysis of protein sequences, namely the computational prediction of the localisation of peroxisome proteins.

Building on existing approaches, we addressed the problem by combining machine learning algorithms with different combinations of protein encodings and embeddings.

We found that the (combination of) deep learning embeddings Seq-Vec [13] and UniRep [12] outperformed classical encodings when applied to sub-peroxisomal classification. Our newly proposed prediction tool In-Pero obtained a (double cross-validated) classification accuracy of 0.92.

We also adopted the approach deployed in In-Pero to predicting the subcellular localisation of mitochondrial proteins, resulting in the In-Mito classifier. We found In-Mito to compare favourably with state-of-the-art approaches and for certain classes of proteins (matrix and intermembrane) to outperform existing prediction tools like DeepMito [34] and SubMitoPred [32].

These results suggest that (*i*) the evolutionary, biochemical and structural information encoded in a protein amino acid sequence cannot be fully captured by one single embedding and that different approaches need to be combined, (ii) deep-learning embeddings are highly versatile and could become a standard for protein sequence representation and analysis and (iii) the possibility of extending In-Pero and In-Mito for the characterisation of other sub-organelles proteins.

Moreover, while in this work we utilised machine learning approaches, we anticipate that our method can be extended to the use of deep-learning methods also for the prediction, such as convolutional neural networks, recurrent neural networks or a combination thereof.

The lack of predictors and tools specifically dedicated to the prediction of sub-localisation of peroxisomal protein makes our work the very first on this subject and presents a complete method and benchmark that can be used as a base for future studies.

## 4. Materials and Methods

### 4.1. Overview of the Full Comparison Workflow

A complete overview of the comparison strategy for the selection of the best classification strategy to predict the sub-localisation of peroxisomal proteins is given in Figure 4. The comparison pipeline consists of three main steps:Data curation: Retrieval of peroxisome protein sequence from UniProt, clustering and filtering.Feature extraction: Transformation of the protein sequences into numerical representations capturing protein characteristics using classical encodings (1HOT, PROP and PSSM) and deep-learning embeddings (UniRep and SeqVec).Full comparison. Double cross-validated assessment of the prediction capability of different combination of machine-learning approaches (Logistic Regression, Support Vector Machines, Partial Least Square Discriminant Analysis and Random Forest) and protein sequence encodings and embeddings using Step Forward Feature Selection.

All methods and approaches used are detailed in the following sections.

### 4.2. Data Sets

Amino acid sequences for peroxisomal membrane and matrix proteins were retrieved in 6 December 2019 from the UniprotKB/SwissProt database (www.uniprot.org) [35].

#### 4.2.1. Retrieval of Peroxisomal Membrane Proteins

Peroxisomal membrane proteins were retrieved using the query ‘fragment:no locations:(location: “Peroxisome membrane [SL-0203]”) AND reviewed:yes’ with peroxisomal membrane sub-cellular location (SL-0203) to select reviewed, non-fragmented membrane protein sequences.

We obtained 327 non-fragmented protein sequences which were then clustered using Cd-hit [36], with sequence identity of 40%. The representative ( i.e., the longest protein sequence in the cluster) of each cluster was chosen resulting in 162 sequences. We used a 40% similarity threshold consistently with DeepMito [15].

We restricted further the selection only to those proteins with at least one associated publication specific for the sub-cellular localization, obtaining 135 highly curated peroxisomal membrane protein sequences. Additionally, three sequences were removed from the data set, since they were not available for the UniRep embedding. The final data set contains 132 membrane proteins.

#### 4.2.2. Retrieval of Peroxisomal Matrix Proteins

Reviewed, non-fragmented matrix peroxisomal protein sequences were obtained with the query ‘fragment:no locations:(location:“Peroxisome matrix [SL-0202]”) AND reviewed: yes’.

We obtained 60 entries that were reduced to 22 after clustering for similarity and further reduced to 19 after selecting only those proteins with at least one publication specific for the subcellular localisation.

Due to the low number of matrix proteins in comparison to the number of membrane proteins (132), we performed another advanced search in Uniprot with query ‘fragment:no locations:(location:“Peroxisome [SL-0204]”) NOT locations:(location:“Peroxisome membrane [SL-0203]”) AND reviewed:yes’ selecting reviewed, non-fragmented protein sequences, with peroxisomal location (SL-0204), and not peroxisomal membrane location (SL-0203).

We obtained 721 non membrane protein sequences, 202 after clustering, which were reduce to 22 after applying the same filtering procedure. There were 13 common entries between the two subsets; clustering using a 40% sequence similarity threshold gave 28 unique peroxisomal matrix protein sequences.

#### 4.2.3. Retrieval of Candidate Peroxisomal Proteins

Further peroxisomal protein candidates were retrieved from Uniprot (1 June 2020). We looked for peroxisomal proteins (SL-0204 and GO:5777) with a non-specific sub-peroxisomal location (SL-0203, SL-0202, GO:5778, GO:5782) and experimental evidence. We then excluded the peroxisomal proteins also found in mitochondria (SL-0173, GO:5739) and in the endoplasmic reticulum (SL-0095, GO:10168), obtaining 116 reviewed entries.

#### 4.2.4. Data Sets for Sub-Mitochondrial Protein Classification

To assess the applicability of our prediction tool to the prediction of other sub-organelles protein localisation we considered two well-curated data sets containing mitochondrial proteins.

*SM424-18 data set*: this data set was used to build the DeepMito predictor [15] and contains 424 mitochondrial proteins collected using stringent conditions, in particular only non-fragmented proteins with an experimentally determined subcellular localisation in one of the four sub-mitochondrial compartments (outer membrane, inter-membrane space, inner membrane and matrix). Clustering using Cd-hit [36], with a 40% sequence identity threshold was used to select representative sequences. We refer the reader to the original publication for more details [15]. *SubMitoPred data set*: this data set was used to build the SubMitoPred predictor [32]. It contains 570 mitochondrial proteins collected using stringent conditions, in particular only non-fragmented proteins with and experimentally determined subcellular localisation in one of the four sub-mitochondrial compartments (outer membrane, inter-membrane space, inner membrane and matrix). Clustering using Cd-hit [36], with a 40% sequence identity threshold was used to select representative sequences. We refer the reader to the original publication for more details [32].

### 4.3. Classic Protein Sequence Encoding Methods

We considered three of the most commonly used method for the encoding of the amino acid protein sequences.

Residue one-hot encoding. The one-hot encoding (1-HOT) [19] is the most used binary encoding method. A residue *j* is represented by a 1×20 vector containing 0 s except in the *j*-th position; for instance alanine (A) is represented as 100,000,000,000,000,000,000. A protein sequence constituted by *L* amino acid is thus represented by an L×20 matrix.Residue physical-chemical properties encoding. Akinori et al. devised a way to represent an amino-acid with ten factors [20] summarising different amino acid physico-chemical properties. This encoding method, often abbreviated as PROP, is the most commonly used physico-chemical encoding [19]. Any given residue *j* in the protein sequence is represented by a 1×10 vector containing real number. Each number summarise different amino-acid properties and it is an orthogonal property obtained after multivariate statistical analysis applied to a starting set of 188 residue-specific physical properties. A protein sequence constituted by *L* amino acid is thus represented by an L×10 matrix.The Position-specific scoring matrix (PSSM) [21,22] takes into account the evolutionary information of a protein. This scoring matrix is at the basis of protein BLAST searches (BLAST and PSI-BLAST) [37] where residues are translated into substitution scores. A residue *j* in the protein sequence is represented by a 1×20 vector containing the 20 specific substitution scores. Amino acid substitution scores are given separately for each position of the protein multiple sequence alignment (MSA) after running PSI-BLAST [37] against the Uniref90 data set (release Oct 2019) for three iterations and e-value threshold set to 0.001. We used a sigmoid function to map the values extracted from the PSI-BLAST checkpoint file in the range [0–1], as in DeepMito [15]. Basically, PSSM captures the conservation pattern in the alignment and summarises evolutionary information of the protein. In PSSM a protein sequence constituted by *L* amino acid is thus represented by an L×20 matrix.

### 4.4. Deep Learning Protein Sequence Embeddings

We considered two recently proposed methods for the embedding of protein sequences based on deep-learning approaches:Unified Representation. The Unified Representation (UniRep) [12] is based on a recurrent neural network architecture (1900-hidden unite) able to capture chemical, biological and evolutionary information encoded in the protein sequence starting from ∼24 million UniRef50 sequences [38]. Technically, the protein sequence is modelled by using a hidden state vector, which is recursively updated based on the previous hidden state vector. This means that the method learns scanning a sequence of amino acids, predicting the next one based on the sequence it has seen so far. Using UniRep a protein sequence can be represented by an embedding of length 64, 256, or 1900 units depending on the neural network architecture used. In this study, we used the 1900 units long (average final hidden array). For a detailed explanation on how to retrieve the UniRep embedding, we refer the reader to the specific GitHub repository:https://github.com/churchlab/UniRep (accessed on 6 June 2021).Sequence-to-Vector embedding. The Sequence-to-Vector embedding (SeqVec) [13] embeds biophysical information of a protein sequence taking a natural language processing approach considering amino acids as words and proteins as sentences. SeqVec is obtained by training ELMo [39], a deep contextualised word representation that models both complex characteristics of word use (e.g., syntax and semantics), and how these uses vary across linguistic contexts, which consists of a 2-layer bidirectional LSTM [40] backbone pre-trained on a large text corpus, in this case, UniRef50 [38]. The SeqVec embedding can be obtained by training ELMo at the per-residue (word-level) and per-protein (sentence-level). With the per-residue level it is possible to obtain a protein sequence embedding that can be use to predict the secondary structure or intrinsically disordered region; with the per-protein level embedding it is possible to predict subcellular localisation and to distinguish membrane-bound vs. water-soluble proteins [13]. Here we use the per-protein level representation, where the protein sequence is represented by an embedding of length 1024. For a detailed explanation on how to retrieve the SeqVec embedding, we refer the reader to the specific GitHub repository: https://github.com/mheinzinger/SeqVec (accessed on 6 June 2021).

### 4.5. Step Forward Feature Selection

Step Forward Feature Selection was used to select the best combination of features (predictors) that is, protein encodings or embeddings to be used as input for classification algorithms [41].

It is a wrapper method that evaluates subsets of variables, in our case, combinations of protein encodings/embeddings. It starts with the evaluation of each individual encoding, and selects that which results in the best performing selected algorithm model. Next, it proceeds by iteratively adding one encoding/embedding to the current best performing features and evaluating the performance of the classification. The procedure is halted when performance worsens and the best combination of embeddings/encodings is retained. A schematic representation of this approach given in Figure 4 (Step 3: Full Comparison).

### 4.6. Classification Algorithms

The determination of the sub-localisation of peroxisomal (membrane vs matrix) protein is easily translated into a two-group classification problem. For this task we considered four widely used machine learning methods (hyperparameters optimisation details in Table 4).

Support Vector Machines (SVM) is an algorithm for two-group classification which aims to find the maximal margin hyperplane separating the points in the feature space [42,43].Random Forest [44,45] is an ensemble learning method that, in the case of a classification task construct a multitude of decision trees and output the mode of the classes of the individual trees.Partial least squares discriminant analysis (PLS-DA) is a partial least squares regression [46,47] where the response vector *Y* contains dummy variables indicating class labels (0–1 in this case). Sample predicted with Y≥0.5 are classified as belonging to class 1 and to class 0 other wise. PLS finds combinations of the original variable maximizing the covariance between the predictor variable and response *Y* by projecting the data in a *k*-dimensional space with *k* possibly much smaller than the original number of variables.Logistic Regression (LR). We used a penalised implementation of multivariable logistic regression [48].

### 4.7. Model Calibration and Validation

We used double cross validation (DCV) [49,50] for (*i*) optimising the hyper-parameters of the different classification algorithm used (i.e., for model calibration) and (ii) for an unbiased estimation of prediction errors when the model is applied to new cases (that are within the population of the data used). This strategy is particularly well suited for small data sets.

The DCV strategy consists of two nested cross-validation loops. In the outer loop data is first split in *k* folds. One fold is used as Validation set while the remaining k−1 folds are used as calibration set. The inner loop is applied to the Calibration set which is again split in a test and training set using *k*-fold split. In our work we used 5 folds for both inner and outer loop. The inner loop is used to optimise the hyperparameters of the different classification algorithms through a (hyper)grid search: for each set of hyperparameters, the average classification score is computed across the folds. The hyperparameters corresponding to the best classification score are then used to fit a classification model whose quality is assessed on the Validation set obtaining unbiased model evaluation since the validation data has not been used to train the classification model.

The step forward feature selection procedure described in Section 4.5 was included in the calibration loop so that model calibration involved also the selection of the best combination (with respect with model predictive ability) of protein sequence encodings and embeddings.

Given the unbalance of the two class of proteins, different weights were applied to the two class. Class weights were considered to be metaparameters and optimised in the inner calibration loop.

### 4.8. Metrics for Model Classification Accuracy

We used several metrics to quantify the quality of the classification models, namely: accuracy (ACC), F1 score [51], balanced accuracy (BACC) [52], Matthews correlation coefficient (MCC) [53]. Formulas are defined as follows:

Accuracy (ACC) that is, the classification error, defined as
(1)$$ACC=\frac{TP+TN}{TP+TN+FP+FN}$$
where: TP is the number of true positives, FP is the number of false positives; TN and FN are the number of true and false negatives, respectively.

The $F_{1}$ score [51]:(2)$$F_{1}=2\times \frac{PPV\times TPR}{PPV+TPR},$$
where PPV is the as positive predicted value (or precision)
(3)$$PPV=\frac{TP}{TP+FP},$$
and TPR is the true positive rate (recall or sensitivity):(4)$$TPR=\frac{TP}{TP+FN}.$$

The $F_{1}$ score is the harmonic mean of recall and precision and varies between 0, if the precision or the recall is 0, and 1 indicating perfect precision and recall.

The balanced accuracy BACC [52]:(5)$$BACC=\frac{TPR+TNR}{2},$$
(6)$$TNR=\frac{TN}{TN+FP}$$
is the true negative rate or specificity. The BACC is an appropriate measure when data is unbalanced and there is no preference for the accurate prediction of one of the two classes.

The Matthews correlation coefficient (MCC) [53]:(7)$$MCC=\frac{TP\times TN-FP\times FN}{\sqrt{\left(TP+FP)(TP+FN)(TN+FP)(TN+FN\right)}},$$

MCC is the correlation coefficient between the true ad predicted class: it is bound between −1 (total disagreement between prediction and observation) and +1 (perfect prediction); 0 indicates no better than random prediction. The MCC is appropriate also in presence of class unbalance [54].

### 4.9. Prediction of Trans-Membrane Proteins

We used TMHMM (Trans-Membrane Hidden Markov Model) for the prediction of trans-membrane proteins [55,56] available at: http://www.cbs.dtu.dk/services/TMHMM/ (accessed on 17 May 2020)

TMHMM returns the most probable location and orientation of trans-membrane helices in the protein sequence, summarised in several output parameters: ExpAA, the expected number of amino acids in transmembrane helices. If this number is larger than 18 it is very likely to be a transmembrane protein; First60, the expected number of amino acids in transmembrane helices in the first 60 amino acids of the protein; PredHel, the number of predicted transmembrane helices. We refer the reader to the original publications for more details.

We used TMHMM to predict the localisation of peroxisomal protein with unknown sub-localisation (see data sets description in Section 4.2.3)

### 4.10. Software

For all classification algorithms we used the implementation available in the scikit-learn python library (version 0.22.1) [57]. We obtained the PLS-DA algorithm by adapting the PLS regression algorithm to perform a regression with a dummy variable. All data sets and codes are available at https://github.com/MarcoAnteghini and at www.systemsbiology.nl.

## Figures and Tables

**Figure 1 ijms-22-06409-f001:**
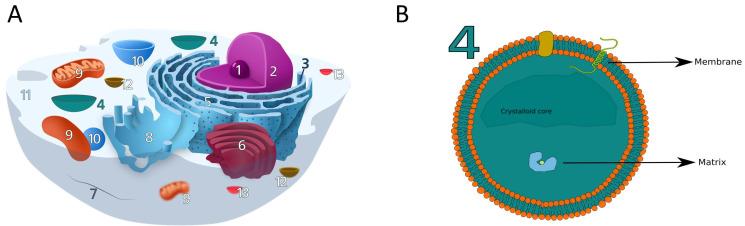
(**A**) The eukaryotic cell and its organelles and compartments: (1) Nucleolus, (2) Nucleus, (3) Ribosome, (4) Peroxisome, (5) Rough Endoplasmic Reticulum, (6) Golgi apparathus, (7) Cytoskeleton, (8) Smooth endoplasmic reticulum, (9) Mitochondrion, (10) Vacuole, (11) Cytoplasm, (12) Lysosome, (13) Vescicles. (**B**) The peroxisome and its structure, showing the lipidic bilayer membrane, the inner matrix and crystalloid core (not always present). Peroxisomal proteins can be divided into two groups, matrix and membrane proteins, depending on the localisation. Membrane proteins are found attached on the inner and outer surface or can span through the layer (trans-membrane proteins). Panel A is partially adapted from en.wikipedia.org/wiki/Ribosome#/media/File:Animal_Cell.svg, accessed on 18 February 2021.

**Figure 2 ijms-22-06409-f002:**
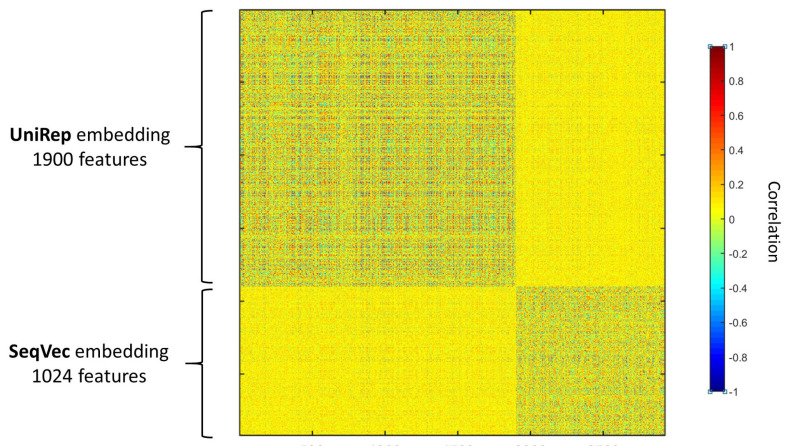
Correlation among the UniRep (1900 features) and the SeqVec (1024 features) protein sequence embeddings. Pearson’s linear correlation is used and are calculated over 160 protein sequences. The two embeddings are uncorrelated.

**Figure 3 ijms-22-06409-f003:**
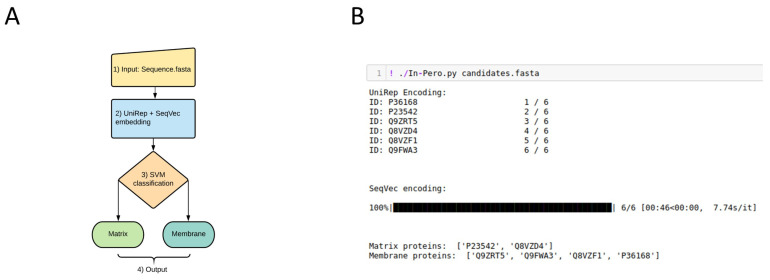
(**A**) In-Pero workflow (1) Input as protein fasta sequence. (2) Sequence representation via DL encoding, in particular concatenating UniRep and SeqVec. (3) Support Vector Machines based classification. (4) Outputof the sub-peroxisomal location of the queried protein. (**B**) Example of a typical execution, with 6 sequences contained in the candidates.fasta file: the sub-peroxisomal classification of each protein is give.

**Figure 4 ijms-22-06409-f004:**
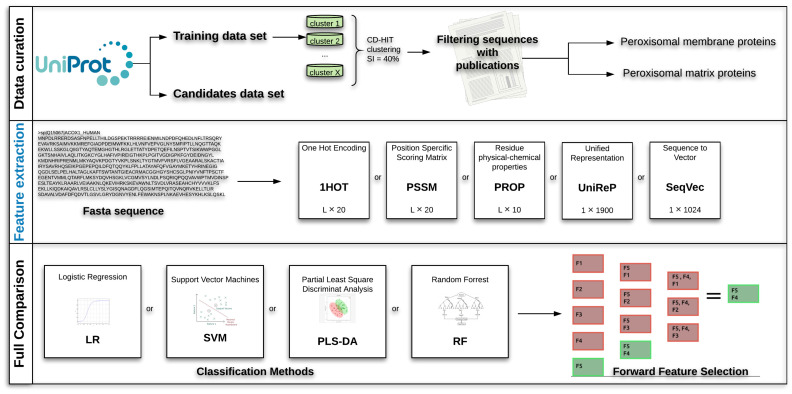
Overview of the full analysis for the predictor pipeline development. Data curation: retrieval and selection of peroxisomal protein sequences (see Section 4.2). Feature extraction: conversion of protein sequences to standard encodings, namely: one-hot encoding (1HOT), residue physical-chemical properties encoding (PROP), position specific scoring matrix (PSSM), unified representation (UniRep), sequence-to-vector (SeqVec). Full Comparison: application of classification algorithms (Section 4.6) and selection of the best combination(s) of sequence encodings and embeddings using step forward feature selection (see Section 4.5).

**Table 1 ijms-22-06409-t001:** Step Forward Feature Selection for each of the compared methods. (**a**) Logistic Regression (LR) performances; (**b**) Support Vector Machines (SVM) performances; (**c**) Partial Least Square–Discriminant Analysis (PLS–DA) performances; (**d**) Random Forest (RF) performances. The analysed encodings and embeddings are protein one hot encoding (1HOT), residue physical-chemical properties encoding (PROP), position specific scoring matrix (PSSM), Unified Representation (UniRep) and Sequence to Vector (SeqVec). The results refer to the Double Cross Validation (DCV) procedure performed for each iteration of the forward feature selection (Section 4.5). The $F_{1}$ (inner) score refers to the inner loop of the DCV while $F_{1}$ (outer) refers to the outer loop. The performances are reported in terms of F1 score, BACC, MCC and ACC (see Section 4.8 and SM).

(**a**) LR					
	$F_{1}$(inner)	$F_{1}$(outer)	BACC	MCC	ACC
1HOT	0.577	0.623 0.071	0.618 0.075	0.269 0.143	0.809 0.036
PROP	0.607	0.595 0.109	0.591 0.093	0.213 0.222	0.794 0.054
PSSM	0.615	0.575 0.067	0.604 0.089	0.177 0.144	0.719 0.040
**UniRep**	**0.765**	**0.749 0.068**	**0.755 0.077**	**0.501 0.137**	**0.856 0.032**
SeqVec	0.792	0.712 0.068	0.726 0.079	0.427 0.140	0.825 0.042
UniRep + 1HOT	0.636	0.648 0.103	0.65 0.111	0.312 0.204	0.806 0.061
UniRep + PROP	0.614	0.595 0.104	0.589 0.093	0.234 0.217	0.812 0.040
UniRep + PSSM	0.634	0.615 0.100	0.615 0.100	0.201 0.166	0.738 0.042
**UniRep + SeqVec**	**0.844**	**0.851 0.055**	**0.847 0.075**	**0.715 0.113**	**0.919 0.032**
(**b**) SVM					
	$F_{1}$(inner)	$F_{1}$(outer)	BACC	MCC	ACC
1HOT	0.624	0.693 0.130	0.713 0.139	0.396 0.261	0.819 0.070
PROP	0.634	0.616 0.108	0.606 0.094	0.274 0.226	0.819 0.041
PSSM	0.631	0.602 0.087	0.623 0.102	0.217 0.178	0.750 0.044
UniRep	0.775	0.768 0.077	0.755 0.099	0.544 0.162	0.869 0.036
**SeqVec**	**0.778**	**0.777 0.046**	**0.813 0.052**	**0.567 0.090**	**0.856 0.038**
SeqVec + 1HOT	0.68	0.757 0.079	0.774 0.103	0.527 0.166	0.856 0.042
SeqVec + PROP	0.648	0.597 0.114	0.589 0.099	0.218 0.229	0.812 0.044
SeqVec + PSSM	0.634	0.614 0.091	0.639 0.110	0.244 0.188	0.756 0.041
**SeqVec + UniRep**	**0.825**	**0.859 0.031**	**0.863 0.042**	**0.721 0.060**	**0.919 0.015**
(**c**) PLS-DA					
	$F_{1}$(inner)	$F_{1}$(outer)	BACC	MCC	ACC
1HOT	0.452	0.452 0.005	0.500 0.001	0.001 0.001	0.825 0.015
PROP	0.551	0.582 0.086	0.575 0.065	0.249 0.198	0.831 0.032
PSSM	0.542	0.592 0.133	0.582 0.092	0.277 0.290	0.844 0.044
**UniRep**	**0.743**	**0.782 0.060**	**0.782 0.060**	**0.568 0.117**	**0.875 0.034**
SeqVec	0.759	0.707 0.081	0.695 0.080	0.419 0.160	0.844 0.044
UniRep + 1HOT	0.478	0.471 0.051	0.502 0.034	0.002 0.112	0.806 0.023
UniRep + PROP	0.478	0.471 0.051	0.502 0.034	0.267 0.128	0.825 0.032
UniRep + PSSM	0.564	0.616 0.110	0.599 0.075	0.326 0.233	0.850 0.041
**UniRep + SeqVec**	**0.806**	**0.792 0.078**	**0.773 0.074**	**0.599 0.166**	**0.888 0.042**
(**d**) RF					
	$F_{1}$(inner)	$F_{1}$(outer)	BACC	MCC	ACC
1HOT	0.569	0.401 0.077	0.523 0.050	0.046 0.089	0.450 0.124
PROP	0.631	0.572 0.016	0.564 0.012	0.203 0.090	0.812 0.020
PSSM	0.618	0.585 0.110	0.567 0.088	0.261 0.261	0.819 0.064
**UniRep**	**0.732**	**0.741 0.051**	**0.779 0.079**	**0.503 0.104**	**0.838 0.023**
SeqVec	0.695	0.691 0.035	0.720 0.053	0.407 0.790	0.800 0.042
UniRep + 1HOT	0.728	0.703 0.063	0.765 0.089	0.443 0.139	0.794 0.032
UniRep + PROP	0.710	0.692 0.093	0.731 0.113	0.403 0.192	0.806 0.041
UniRep + PSSM	0.699	0.743 0.100	0.776 0.128	0.501 0.209	0.844 0.052
**UniRep + SeqVec**	**0.778**	**0.764 0.135**	**0.790 0.141**	**0.540 0.267**	**0.850 0.087**
UniRep + SeqVec + 1HOT	0.774	0.721 0.108	0.738 0.121	0.456 0.214	0.844 0.044
**UniRep + SeqVec + PROP**	**0.720**	**0.787 0.134**	**0.793 0.144**	**0.581 0.261**	**0.888 0.061**
UniRep + SeqVec + PSSM	0.741	0.733 0.123	0.754 0.136	0.480 0.242	0.850 0.054

**Table 2 ijms-22-06409-t002:** Transmembrane and Membrane proteins found with both methods (In-Pero and TMHMM). The seven Transmembrane proteins showing high prediction scores with TMHMM are in bold.

Membrane		Transmembrane
O82399	Q8H191	**O64883**
P36168	Q8K459	**P20138**
P90551	Q8VZF1	**Q84P23**
Q12524	Q9LYT1	**Q84P17**
Q4WR83	Q9NKW1	**Q84P21**
Q75LJ4	Q9S9W2	**Q9M0X9**
Q9SKX5		**P08659**

**Table 3 ijms-22-06409-t003:** Comparison with DeepMito and DeepPred-SubMito (DP-SM) based on the SM424-18 data set (Data A) and the SubMitoPred data set (Data B). The results are reported in terms of Matthews Correlation Coefficient (MCC). The four mitochondrial compartments are outer membrane (O), inner membrane (I), intermembrane space (T) and matrix (M). Given the similar performances of our predictor (In-Mito) implemented with Logistic Regression (LR) and Support Vector Machines (SVM), we report both. The best performances are highlighted in bold font.

Data A	MCC(O)	MCC(I)	MCC(T)	MCC(M)
DeepMito	0.460	0.470	0.530	0.650
DP-SM	**0.850**	0.490	**0.990**	0.560
In-Mito (LR)	0.680	**0.730**	0.690	**0.820**
In-Mito (SVM)	0.640	0.690	0.620	0.800
**Data B**				
SubMitoPred	0.420	0.340	0.190	0.510
DeepMito	0.450	0.680	0.540	0.790
DP-SM	**0.920**	0.690	**0.970**	0.730
In-Mito (LR)	0.690	0.750	0.620	**0.850**
In-Mito (SVM)	0.650	**0.760**	0.540	0.840

**Table 4 ijms-22-06409-t004:** Hyperparameters for the grid searches.

	Hyperparameters
SVM	C:logspace(−2,10,13) gamma:logspace(−9,3,13) kernel:[‘linear’,‘poly’,‘rbf’,‘sigmoid’]
RF	n_estimators:[15,25,50,75,100,200,300] criterion:[‘gini’,‘entropy’] max_depth:[2,5,10,None] min_samples_split:[2,4,8,10] max_features:[‘sqtr’,‘auto’,‘log2’]
PLS-DA	n_components:[2,5,10,15,20,25,30]
LR	penalty:[‘l1’,‘l2’] solver:[‘liblinear’,‘saga’] C:logspace(−3,9,13)

## Data Availability

All presented tools and data are free to use and available online. The data sets used in this study are available at https://github.com/MarcoAnteghini/In-Pero/tree/master/Dataset. Standalone versions of In-Pero and In-Mito are available at https://github.com/MarcoAnteghini/In-Pero and https://github.com/MarcoAnteghini/In-Mito (accessed on 6 June 2021). The data set containing the 116 investigated proteins as candidates and the related prediction is available for experimental classification and proper UniProt annotations at https://github.com/MarcoAnteghini/In-Pero/tree/master/Candidates.

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
