# Peer review of "In-Pero: Exploiting Deep Learning Embeddings of Protein Sequences to Predict the Localisation of Peroxisomal Proteins"

_ijms, 2021, doi:10.3390/ijms22126409_

Round 1

Reviewer 1 Report

Anteghini et al. have developed and evaluated an ML algorithm for the prediction of sub-peroxisomal localization of proteins given their sequence. Given the multifacet role of peroxisomal proteins in normal and pathological cell physiology and, hence, the involvement of their aberrant localization into the pathogenesis of some diseases, it is important to be able to predict their sub-organelle localization. The authors assemble the training set for such a task, evaluate different protein descriptors and their combinations, as well as assess the accuracy of various ML algorithms. The resulting model is made freely available. Moreover, the authors show that similar models can potentially have broader applicability, e.g., for the prediction of sub-mitochondrial localization.

The reviewer has just a couple of comments:

  1. The main focus of the present study is to predict sub-organelle localization. However, little attention is paid to the more general problem of cell-level localization. I.e., in the reviewer’s opinion, it is also important to discuss the state-of-the-art approaches aiming at the prediction of subcellular localization in general and particularly peroxisomal localization.
  2. It would be interesting to compare the performance of the present model simply with the existing approaches for membrane protein prediction. Given that the authors classify peroxisomal proteins into two groups, i.e., membrane and matrix, it is also important to show that their method outperforms the existing predictors of membrane proteins.

Reviewer 2 Report

In this study, Anteghini et al. proposed a predictor for the localization of peroxisomal proteins. Curated data has been manually collected to support training the prediction model. Different machine learning models and feature extraction methods have been assessed to get the optimal model. The idea is of interest, however, there are some major points that need to be addressed:

1. The embedding methods are mostly based on deep learning models, why did the authors not use deep learning to learn such kinds of features rather than conventional machine learning?

2. Sequence similarity of 40% is a high cut-off level. Normally, this field accepts a level lower than 30% to ensure an unbias model.

3. Even with the use of cross-validation strategy, it is important to have an independent dataset to evaluate the prediction model on unseen data.

4. There are different measurement metrics, why did the authors only use MCC in table 3?

5. For this kind of study, the authors should develop a web server to support users submit their sequences to test.

6. Machine learning-based bioinformatics problem is well-known and has been proposed in previous studies i.e., PMID: 33036150 and PMID: 31920706. Therefore, the authors are suggested to refer to more works in this description to attract a broader readership.

7. It is still unclear how did the authors extract deep learning-based features i.e., UniRep, SeqVec, ...

Round 2

Reviewer 2 Report

My previous comments have been addressed well.